# Propagation of Mitochondria-Derived Reactive Oxygen Species within the *Dipodascus magnusii* Cells

**DOI:** 10.3390/antiox10010120

**Published:** 2021-01-15

**Authors:** Anton G. Rogov, Tatiana N. Goleva, Khoren K. Epremyan, Igor I. Kireev, Renata A. Zvyagilskaya

**Affiliations:** 1Bach Institute of Biochemistry, Federal Research Center “Fundamentals of Biotechnology” of the Russian Academy of Sciences 33, bld. 2 Leninsky Ave., Moscow 119071, Russia; lloss@rambler.ru (A.G.R.); goleva13@yandex.ru (T.N.G.); 7700077@mail.ru (K.K.E.); 2Belozersky Institute of Physico-Chemical Biology, Lomonosov Moscow State University, Vorobyevy Gory 1, Moscow 119992, Russia; iikireev@genebee.msu.su

**Keywords:** yeast, oxidative stress, reactive oxygen species, mitochondrial fragmentation, SkQ1

## Abstract

Mitochondria are considered to be the main source of reactive oxygen species (ROS) in the cell. It was shown that in cardiac myocytes exposed to excessive oxidative stress, ROS-induced ROS release is triggered. However, cardiac myocytes have a network of densely packed organelles that do not move, which is not typical for the majority of eukaryotic cells. The purpose of this study was to trace the spatiotemporal development (propagation) of prooxidant-induced oxidative stress and its interplay with mitochondrial dynamics. We used *Dipodascus magnusii* yeast cells as a model, as they have advantages over other models, including a uniquely large size, mitochondria that are easy to visualize and freely moving, an ability to vigorously grow on well-defined low-cost substrates, and high responsibility. It was shown that prooxidant-induced oxidative stress was initiated in mitochondria, far preceding the appearance of generalized oxidative stress in the whole cell. For yeasts, these findings were obtained for the first time. Preincubation of yeast cells with SkQ1, a mitochondria-addressed antioxidant, substantially diminished production of mitochondrial ROS, while only slightly alleviating the generalized oxidative stress. This was expected, but had not yet been shown. Importantly, mitochondrial fragmentation was found to be primarily induced by mitochondrial ROS preceding the generalized oxidative stress development.

## 1. Introduction

Reactive oxygen species (ROS) play a dual role in biological systems. Initially, ROS were described as simple byproducts of metabolism that caused oxidative damage and chronic diseases, as well as aging [1]. However, an increasing amount of evidence has shown that ROS play a more complex role than previously anticipated, serving as critical mediators in cell signaling. As signaling molecules, ROS are highly versatile due to their diverse properties, which include different levels of reactivity, sites of production (intracellular compartmentation), and potential to cross biological membranes. They were found to regulate division, development, differentiation, redox levels, systemic responses, adaptation to stress, interactions with other organisms, and cell death in most eukaryotic organisms [2,3,4]. More specifically, ROS have an impact on cell-signaling proteins, ion channels, and transporters, and modify kinases and the ubiquitination–proteasome system [5].

A fine balance exists between ROS production for signaling, the baseline of metabolically produced ROS, the rate of ROS diffusion and reactivity, and ROS removal and ROS perception in the different compartments of the cell (the ROS network). The integration of these different ROS-dependent reactions/signals determines the overall response of the cell to a particular stimulus [4]. Multiple intracellular ROS generators can be activated, either concomitantly or sequentially, by relevant signaling molecules for redox biological functions [3].

Mild redox stresses can protect an organism from subsequent larger stresses and improve metabolism and immune system, thus improving its health and prolonging its lifespan [6,7]. Furthermore, it has recently been demonstrated that several organisms have extended lifespans when their mitochondrial ROS production is high even for short periods of time [8]. To highlight the positive effects of ROS on cells, the new terms “oxidative eustress” and “oxidative distress” were introduced as an extension of the initial definition of “oxidative stress” [9].

Excessive production of ROS (oxidative stress), as a result of an imbalance between the generation and clearance of ROS, often results in multifarious impairment to cells, including a decrease in the ATP level; elevation of cytosolic Ca^2+^; damage to cell components; including DNA, proteins, and lipids [10]; and a compromise of multiple cellular signaling pathways, thereby leading to the aggravation of many diseases [11,12] and apoptosis induction [13].

It is commonly believed that mitochondria are the initial and principal organelle responsible for ROS generation in the cell, although some enzymes and processes also contribute to ROS generation.

Mitochondria contribute to a wide range of cellular processes, and they are critical for cell survival and death. Apart from their best-known function, which is to meet acute and chronic energetic needs through the generation of ATP, they are fully integrated into the cellular metabolism required for growth and proliferation [14], maintaining calcium homeostasis [15,16] and calcium signaling, and regulating cell adaption to external stressors [7,17]. They are implicated in the precise immune response of the cell [18], modulation of ROS signaling and maintaining oxidative homeostasis [19,20], regulation of apoptosis by sequestering cytochrome c under normal healthy conditions, and orchestrating its release upon stress conditions [21,22,23]. Recently, findings have emerged implicating a fundamental role for mitochondria even in highly glycolytic cells with relatively low energy requirements, and mitochondrial content such as stem cells [24], epithelium [25] and neutrophils [26]. These cell types traditionally have been thought to exhibit little or no mitochondrial functions, with the exception of ROS production, thus emphasizing the critical role of this mitochondrial function in maintaining cellular homeostasis.

Mitochondria possess a number of processes to produce free radicals, with the respiratory chain being a major ROS producer under resting conditions. There is a view that excessive ROS production is largely derived from mitochondrial dysfunction, giving rise to pathogenicity [27,28,29]. Mitochondrial ROS (mROS) are a critical factor in the mechanism underlying muscle atrophy and frailty (i.e., sarcopenia) [30], and they possibly play a role in the pathogenesis of multiple sclerosis [31], in age-related chronic wounds [32], in obesity-related type 2 diabetes [33], and in the aging of mtDNA mutator mice [34]. It has been hypothesized that in long-lived neurons, where active mitochondria function must be maintained for an entire lifetime, the role of mitochondrial ROS in normal function and pathology (Alzheimer’s, Parkinson’s, motor-neuron and Huntington’s diseases) may be crucial [35]; in cancer cells, mROS production contributes to evolution toward more-aggressive phenotypes [36]. Atrophy of thymic epithelial cells is also impacted by mROS [37].

However, it is worth noting that conclusions on the possible involvement of mROS in the development of the pathologies mentioned above were primarily based on the ability of mitochondria-targeted antioxidants to ameliorate or even prevent the pathological effects [29,38,39,40,41,42,43,44,45,46,47]. All of these mitochondrial-targeted agents have contributed to a novel area of therapy called mitotherapy [48].

The mechanisms underlying propagation and progression of oxidative stress within the cell have been the focus of researchers’ interests for more than 15 years. It was suggested and then found that exposure to excessive oxidative stress triggers ROS-induced ROS release (RIRR) in neighboring mitochondria and the release of generated ROS into cytosol. This mitochondrion-to-mitochondrion ROS-signaling can constitute a positive feedback mechanism for enhanced ROS production [49,50,51]. Several propagation models of RIRR have been put forward that implicate mitochondrial carriers [5,49,51,52,53,54,55,56,57] (for more details, see Discussion). However, most of this research was carried out on cardiac myocytes, rather than specific cells, in which the distribution and metabolic function of the mitochondria varied depending on the developmental stage of the myocardium [58]. Only in neonatal cardiac myocytes do mitochondria exhibit a characteristic reticular distribution in the cytosol that allows them to move freely. In adult cardiac myocytes, a large portion of the mitochondria are tightly packed between myofibrils to form a cell-wide network of communicating organelles, accounting for about 35% of the myocytes’ volume [59]. The lattice-like arrangement of mitochondria, mostly in long and dense rows parallel to the cardiac myocyte myofilaments, has the structural features of a highly ordered network [60]. There has been skepticism regarding whether this subpopulation of mitochondria would actually experience fission and fusion, as well as mitochondrial mobility [55,61,62]. However, there is growing evidence, primarily from loss-of-function studies of proteins involved in these processes, that mitochondrial morphodynamics could play a role in the maintenance of mitochondrial fitness [55,63]. It may be that in complex cells such as cardiac myocytes, other functions of the fission and fusion proteins are more important than mitodynamics, such as mitophagy, which can remove damaged mitochondria without ongoing repetitive fission and fusion [62].

From the above, we believe that cardiac myocytes are not the best models for studying the cross-talk between propagation of oxidative stress within the cell and mitochondrial dynamics.

The main goal of this study is to follow propagation of oxidative stress in the cell and its cross-talk with mitochondrial dynamics (fragmentation of mitochondria). To this end, we used the nontoxic ascomycetous yeast *Dipodascus magnusii* (from the Latin meaning “giant”) as a model organism. It is pertinent to note that yeasts and humans, despite their evolutionary divergence that resulted in dramatic differences in cell and tissue organization, motility, and environment [64], share well-conserved molecular and cellular mechanisms of eukaryotic cell biology. These include commonalities in their underlying molecular makeup, the universal mechanisms that govern signaling pathways [65,66], protein folding, quality control and degradation, mitochondrial dysfunction, oxidative stress, a secretory pathway, vesicular trafficking [67,68,69], and even mechanisms of cell death and survival [68,70]. Moreover, numerous processes and mechanisms, such as cell signaling pathways that regulate metabolism, cell growth and division, organelle functions, cellular homeostasis, stress responses, and mitochondrial dynamics (fission and fusion) [71], were first identified in yeasts and then shown to be conserved in higher eukaryotes. Because of the high complexity of human cells, yeasts, which are simple unicellular eukaryotic organisms, have become a valuable and prevalent eukaryotic model for deciphering human biology and pathologies thanks to their inherent tractability, ease of genetic manipulation, and high-throughput screening technologies [72,73,74].

*D. magnusii* cells have multiple advantages over other models. They are unique in that they contain a respiratory metabolism closely resembling that of mammalian cells [75,76]. In contrast to cell cultures, they vigorously grow on a variety of simple, well-defined and inexpensive media, and respond quickly to various substances. Additionally, these giant cells normally contain freely moving mitochondria that are fragmented when oxidative stress is imposed [77,78,79,80]. If we add easy-to-visualize mitochondria in the cell and the possibility to explore a large number of individual cells using modern methods of flow cytometry and time-lapse microscopy, it becomes clear to us that *D. magnusii* cells are particularly suitable for revealing a cross-talk among oxidative stress, its propagation, and mitochondrial dynamics.

Using a combination of fluorescence time-lapse microscopy and flow cytometry assays, we followed the spatiotemporal development of prooxidant-induced oxidative stress in mitochondria and in protoplasm. We found that oxidative stress detectable in mitochondria with MitoSox Red increased immediately after the cell treatment and preceded occurrence (emergence) of generalized oxidative stress in the whole cell. Moreover, progressively increased oxidative stress in mitochondria was accompanied by gradual mitochondrial fragmentation.

## 2. Materials and Methods

### 2.1. Chemical Reagents

The Amplex Red, catalase from bovine liver, peroxidase from horseradish, PMSF, tert-butyl hydroperoxide, and Tris were purchased from Sigma-Aldrich Ltd. (St. Louis, MO, USA); the bacto agar, peptone, and yeast extract were purchased from Becton, Dickinson and Company (Franklin Lakes, NJ, USA); the bovine serum albumin and glycerol were purchased from MP Biomedicals (Santa Ana, CA, USA); the H_2_DCF-DA, Hoechst-33342, MitoSox Red and Mitotracker Green were purchased from Life Technologies (Carlsbad, CA, USA); the CaCl_2_, KCl, KH_2_PO_4_, and K_2_HPO_4_ were purchased from Merck (Kenilworth, NJ, USA); and the Amplite™ Colorimetric Superoxide Dismutase (SOD) Assay Kit was purchased from AAT Bioquest (Sunnyvale, CA, USA). Other reagents of the highest available quality were obtained from domestic suppliers. The SkQ1 and MitoCLox were kindly provided by Dr. K.G. Lyamzaev of the Belozersky Institute of Physico-Chemical Biology.

### 2.2. Cell Culture

The VKM Y261 strain of the *Dipodascus magnusii* yeast, from an all-Russian collection of microorganisms was cultivated at 28 °C in agitated (220 rpm) 750-mL Erlenmeyer flasks in 100 mL of a semi-synthetic medium [77] containing 1% glycerol as the source of carbon and energy. The cells were harvested at the exponential growth phase (OD590 ~ 1.0).

### 2.3. Cell Viability Assay

Cell viability was detected with propidium iodide (PI), a commonly used fluorescent reagent that cannot pass through cell membranes of living cells, but is able to enter necrotic and dead cells by binding nonspecifically to nucleic acids [81]. The fluorescence of the PI was measured by flow cytometry using a FACSCalibur flow cytometer (Becton, Dickinson and Company, Franklin Lakes, NJ, USA). The data obtained for 15,000 cells were stored and analyzed on a logarithmic scale using FlowJo Software v10.5 (FlowJo LLC, Ashland, OR, USA).

### 2.4. ROS Generation and Determination

Oxidative stress in the yeast cells was induced by *tert*-butyl hydroperoxide (*t*-BHP), a well-known model prooxidant causing ROS formation through enhanced lipid peroxidation reaction [82]. The ROS generation in yeast cells was determined using H_2_DCF-DA and MitoSox Red dyes. The membrane-permeable nonfluorescent H_2_DCF-DA enters yeast cells as a diacetate, wherein acetyl groups are cleaved by intracellular esterases, thus converting the dye to a membrane-impermeable form of DCF capable of emitting fluorescence upon oxidation, primarily due to hydrogen peroxide. This makes DCF suitable for detecting oxidative stress in the cytoplasm. Under correct conditions, MitoSox Red is a probe for the mitochondrial superoxide anion radical, triggering emission of red fluorescence [83]. However, MitoSox Red, depending on the dose used, can change its localization within the cell and nonspecifically stain DNA [84,85]. Therefore, the optimal MitoSox Red concentration was determined in special preliminary experiments. For DCF, some complications are also known, such as oxidation of H_2_DCF catalyzed by exogenous hemin or heme-containing proteins [86], or by cytochrome c, which is released from mitochondria during cell death [87]. Therefore, the incubation time was limited to 80–120 min, when the cells remained viable (for details, see Results).

Yeast cells harvested at the exponential growth phase were washed with 50 mM PBS, pH 5.5, and stained with 15 µM H_2_DCF-DA and 1 µM MitoSox Red for 30 min. Stained cells were washed in a fresh portion of incubation medium and exposed to 750 µM *t*-BHP. To avoid photooxidation, all incubations were performed in the dark. Fluorescence of DCF and MitoSox Red was measured by flow cytometry with a FACSCalibur flow cytometer (Becton, Dickinson and Company, Franklin Lakes, NJ, USA). The data obtained for 15,000 cells were stored and analyzed on a logarithmic scale using FlowJo Software v10.5 (FlowJo LLC, Ashland, OR, USA).

### 2.5. Visualization of Mitochondria in Cells

For visualization of mitochondria, yeast cells were stained with the fluorescent probe MitoTracker Green FM, which labels mitochondria in a membrane potential-independent manner [88]. Stained cells were analyzed under an Axioscop 40 fluorescence microscope (Zeiss, Oberkochen, Germany). Image processing was performed using Icy software 2.1.0.1 [89].

### 2.6. Time-Lapse Microscopy

For time-lapse microscopy, *D. magnusii* cells harvested at the exponential growth phase were stained as described above, plated on 24-well microscopy plate covered with Concanavalin A (for solid-phase immobilization of cells on glass), and exposed to 750 µM *t*-BHP at room temperature. Images were acquired at 3-min intervals with a light-exposure duration of 100 ms and 250 ms for 488 nm and 561 nm excitation, respectively. Imaging was performed using an inverted motorized Eclipse Ti-E microscope with a PerfectFocus autofocusing system (Nikon, Melville, NY, USA). The microscopy system was equipped with a 100x Apo TIRF Oil objective (NA1.49) and cooled EM-CCD camera iXonDU-897E (Andor Technology, Belfast, UK) under the control of NIS-Elements 4.0 software. Post-processing was performed using Icy software 2.1.0.1 [89].

### 2.7. Multicolor Staining

For multicolor staining, yeast cells were stained with three fluorescent probes, including H_2_DCF-DA, MitoSox Red, and Hoechst-33342 (as a marker for nuclei). Cells harvested at the exponential growth phase were rinsed with 50 mM PBS, pH 5.5 and stained with 15 µM H_2_DCF-DA, 1 µM MitoSox Red, and 5 µg/mL Hoechst-33342 for 30 min [90]. Stained cells were washed in a fresh portion of incubation medium and exposed to 750 µM *t*-BHP. Every 20 min, a sample of stained cells was observed under the Axioscop 40 fluorescence microscope (Zeiss, Oberkochen, Germany). Image processing was performed using Icy software 2.1.0.1 [89].

### 2.8. Mitochondrial Lipid Peroxidation Assay

Mitochondria lipid peroxidation was measured with the novel mitochondria-targeted dye MitoCLox [91,92]. *D. magnusii* cells were washed in 50 mM PBS, pH 5.5, incubated with 100 nM MitoCLox for 30 min, then washed in a fresh portion of growth medium and incubated with 800 nM SkQ1 for 1 h. Cells were then washed with a fresh portion of growth medium and incubated with 750 µM *t*-BHP. Flow cytometry analyses were performed every 10 min using a FACSCalibur flow cytometer (Becton, Dickinson and Company, Franklin Lakes, NJ, USA) at FL1 channel (530/30 nm). The data obtained for 15,000 cells were stored and analyzed on a logarithmic scale using FlowJo Software v10.5 (FlowJo LLC, Ashland, OR, USA).

### 2.9. Preparation of Cellular Homogenate

The *D. magnusii* cells were washed twice with 50 mM PBS, pH 5.5, suspended in the assay buffer, and disrupted with Tissuelyser LT (Qiagen, Hilden, Germany) for 2 min at 50 Hz and 0–4 °C. The obtained homogenate was centrifuged at 10,000× *g* for 30 min at 4 °C, and 0.1 mg protein of the resulting supernatant was used for determination of SODs activity. Cellular proteins were assayed by the Bradford method [93], using BSA as a standard.

### 2.10. Superoxide Dismutase Activity Assay

SODs activity was determined at 25 °C using an Amplite™ Colorimetric Superoxide Dismutase (SOD) Assay Kit (AAT Bioquest, Sunnyvale, CA, USA). A SOD-mediated decrease in an absorption of the product of ReadiView™ SOD560 oxidation by superoxide anion in the xanthine/xanthine oxidase mixture was measured at 560 nm using a Cary 300 Bio spectrophotometer (Varian Medical Systems, Palo Alto, CA, USA) [94]. The results were expressed as SODs units per mg of cell protein.

### 2.11. Statistical Analysis

All experiments were performed at least three times, with consistent results. To assess mitochondrial morphology, at least 100 individual cells were examined in each trial. Statistical evaluations were performed using the one-way ANOVA test with the Posthoc Tukey HSD test. Data are presented as mean ± S.E. from at least three independent replicates.

## 3. Results

### 3.1. Propagation of ROS Production in Yeast Cells

One of the main goals of the work was to trace the development of oxidative stress in yeast cells initiated by the prooxidant *t*-BHP. Giant *D. magnusii* cells with their highly structured mitochondrial reticulum that collapses and turns into multiple small (fragmented) mitochondrial structures under imposed oxidative stress (Figure 1), are especially suitable for this kind of research.

To this end, we loaded *D. magnusii* cells with both MitoSox Red (to detect mitochondrial ROS) and H_2_DCF-DA (to track ROS in the entire cell), then exposed them to the prooxidant *t*-BHP, and analyzed the spatiotemporal development (propagation) of oxidative stress.

In specially designed control experiments, the optimal concentrations of MitoSox Red (1 μM), H_2_DCF-DA (15 μM), *t*-BHP (250 or 750 μM) and optimal permissible duration of experiments were empirically chosen. The fluorescence of oxidized MitoSox Red in all experiments was assessed only for an 80-min time period, during which no co-localization of Hoechst-33342 (a marker for nuclei) and MitoSox Red in cells subjected to oxidative stress was observed (Figure 2), thus ensuring that under the used optimal experimental conditions, MitoSox Red at a concentration of 1 μM selectively detected mitochondrial oxidative stress.

Exponentially growing *D. magnusii* cells washed by a fresh portion of the growth medium were loaded with 1 µM MitoSox Red and 15 µM H_2_DCF-DA for 30 min, washed by 50 mM PBS, pH 5.5, and then incubated with 750 µM *t*-BHP in the dark to avoid excessive photoinactivation of MitoSox Red and autoactivation of DCF. The stained cells were examined using two independent methods: flow cytometry and time-lapse microscopy.

It was found that in the untreated (control) cells, DCF fluorescence remained negligible throughout the experiment, while MitoSox Red fluorescence (an indicator of mitochondrial ROS) remained at a low but distinct level, possibly reflecting the basal ROS production by the respiratory chain (Figure 3a). In cells exposed to the prooxidant (Figure 3b), the oxidative stress induced by *t*-BHP initially developed only in mitochondria, starting almost immediately after contact with the prooxidant, and then progressively increased. Importantly, ROS production in mitochondria assayed with MitoSox Red far preceded the appearance of the generalized oxidative stress in the yeast cell assessed with H_2_DCF-DA. Once started, the generalized oxidative stress also exponentially increased, but after a prolonged lag period. Similar results were obtained using time-lapse fluorescence imaging (Figure 4).

Progression of the mitochondrial oxidative stress was independently verified by measuring mitochondrial lipid peroxidation using the novel mitochondria-addressed dye MitoCLox [91,92] (Appendix A). Notably, in conformity with the data presented in Figure 3 and Figure 4, an increment in mitochondrial lipid peroxidation occurred predominantly before the development of the generalized oxidative stress.

It is known that oxidative stress is accompanied by the activation of enzymes of antioxidant defense. It was found that 1-h exposure of yeast cells to *t*-BHP, coinciding with the development primarily of mitochondrial oxidative stress, did not lead to a significant increase in SODs activity, while prominent SODs activation was seen after 2 h of prooxidant exposure (Appendix A), possibly reflecting propagation of oxidative stress. For mitigation of oxidative stress, we used SkQ1 (10(6′plastoquinonyl)decyltriphenylphosphonium), the positively charged mitochondria-targeted antioxidant, a conjugate of lipophilic cation triphenylphosphonium with ubiquinone, a component of the photosynthetic electron transport chain [95]. The data obtained for planar phospholipid membranes, liposomes, isolated mitochondria, and cells showed that SkQ1 is a very efficient antioxidant acting at low concentrations [96,97]. It has been widely applied in treating a large number of pathologies related to oxidative stress [29]. Previously, we also showed that SkQ1 alleviated and even prevented prooxidant-induced oxidative stress in yeasts [77,79,80].

It was found that preincubation of yeast cells with SkQ1 substantially diminished ROS production by mitochondria (Figure 5), while only slightly alleviating the generalized oxidative stress (Figure 6).

### 3.2. Mitochondria Fragmentation in Yeast Cells

The large cells with well-structured mitochondrial reticulum of *D*. *magnusii* allowed us to trace the relationship between the development of both mitochondrial and generalized oxidative stresses and mitochondrial structure. In the control samples, the mitochondrial reticulum remained intact throughout the experiment (Appendix A). Incubation with the prooxidant was accompanied not only by a gradual increase in fluorescence of MitoSox Red, but also by gradual fragmentation of mitochondria (Figure 4, Appendix A).

To establish close correlations between mitochondrial fragmentation and the development of mitochondrial or generalized oxidative stresses (i.e., ROS detected by MitoSox Red or DCF, respectively), we analyzed the moments of complete mitochondrial fragmentation, as well as the apparent time points of the onset of DCF fluorescence changes in each of 101 individual yeast cells (Figure 7). Fragmentation was considered complete when all mitochondrial structures in the cell were represented by scattered balls or ellipses, and no filamentary or branched structures were subsequently formed.

An increase in DCF fluorescence in all analyzed cells occurred strictly after the mitochondrial fragmentation peak had passed (Figure 7), suggesting that mitochondrial fragmentation was primarily induced by mitochondrial ROS (mROS), and was an event preceding the generalized oxidative stress development. SkQ1 did not reduce mitochondrial fragmentation.

Four main conclusions can be drawn from the presented results. First, the *t*-BHP-induced oxidative stress initially developed only in mitochondria, beginning almost immediately after contact with the prooxidant, and then progressively increased. Secondly, ROS production in mitochondria (assayed with MitoSox Red) far preceded the appearance of the generalized oxidative stress (assessed with H_2_DCF-DA) in the yeast cell; once started, the generalized oxidative stress was increased, but after a prolonged lag period. Third, the preincubation of yeast cells with the mitochondria-targeted antioxidant SkQ1 substantially diminished ROS production by mitochondria, while only slightly alleviating the generalized oxidative stress (Figure 5 and Figure 6). Finally, mitochondrial fragmentation was primarily induced by mROS, preceding the development of the generalized oxidative stress.

## 4. Discussion

Mitochondrial ROS (mROS) can be integral to cell survival, but can also be deleterious, a phenomenon defined as mitohormesis [6,7,17]. Therefore, it is not surprising that mROS and their evolution are currently subjects of intense scrutiny by the scientific community.

There is a consensus that mROS, primarily the superoxide anion radical, is the first form of ROS to propagate [49,57]. It was found that the exposure to excessive oxidative stress resulted in triggering ROS-induced ROS release (RIRR) in neighboring mitochondria and the release of generated ROS into cytosol [51]. Most of these studies have been carried out on cardiac myocytes, with their mitochondrion consisting of a lattice-like, cell-wide network of densely packed organelles [56,60]. This specific spatiotemporal organization of mitochondria in cardiac myocytes facilitates myocyte-wide, cluster-bound, mitochondrial inner membrane potential oscillatory depolarizations, commonly triggered by metabolic or oxidative stressors. The association of oscillations of the inner membrane potential with ROS was discovered by Zorov and Sollot, who showed that a localized laser flash produced a large number of free radicals in cardiac mitochondria, concomitant with depolarization, referred to as RIRR [49,51]. Local intermitochondrial coupling mediated by reactive oxygen species (ROS) can activate inner membrane pores to initiate a ROS-induced-ROS-release process that produces synchronized limit cycle oscillations of mitochondrial clusters within the whole mitochondrial network [55,56]. A functional link between metabolic oscillations and cardiac function was demonstrated by O’Rourke et al. [98]. The cyclical activation of ATP-sensitive potassium currents in guinea pig cardiac myocytes under substrate deprivation was associated with low frequency oscillations in the myocyte’s action-potential duration and excitation-contraction coupling. These oscillations were accompanied by a synchronous oxidation and reduction of the intra-cellular nicotinamide adenine dinucleotide (NADH) concentration. RIRR has been shown to underlie propagated waves of membrane depolarization, as well as synchronized limit cycle oscillations of membrane potential in the network. This mechanism is controlled by ROS scavenging processes so that, in a confined environment, a cyclical activation of RIRR can lead to local mitochondrial membrane potential oscillations [57]. Several propagation models for RIRR have been put forward that implicate mitochondrial carriers [5,49,51,52,53,54,55,56,57]. Under conditions that lead to RIRR, the increase in ROS reaches a threshold level that triggers the opening of one of the requisite mitochondrial channels, such as the mitochondrial permeability transition pore (mPTP) [49,51], inner membrane anion channel (IMAC) [52,53,54,55,56], or ATP-depended K-channel [99]. Very recently, some details have emerged showing that superoxide mediates activation of energy-dissipating ion channels, while hydrogen peroxide is an important communicator of oxidative stress, distributing stress throughout the network prior to any signs of network failure from depolarizations of the mitochondrial membrane potential [57].

From the above, it is clear to us that cardiac myocytes, with their specific lattice-like arrangement of mitochondria, are not typical of the absolute majority of eukaryotic cells. With no mitochondrial motility, fission and fusion is not an adequate model for revealing rather complex cross-talk between oxidative stress and mitochondrial dynamics. Additionally, in this research, the main emphasis was placed only on propagation of the mitochondrial membrane depolarization.

The aim of this work was to trace the spatiotemporal development of oxidative stress in yeast cells initiated by the prooxidant *t*-BHP, and to follow fragmentation of mitochondria in individual cells as a result of this oxidative stress propagation. Using the *D*. *magnusii* yeast in a combination of fluorescence microscopy time-lapse microscopy and flow cytometry assays, we gained a novel insight into the propagation of oxidative stress within the cell. It was shown that under optimized experimental conditions, the oxidative stress induced by *t*-BHP initially was developed only in the mitochondria, beginning almost immediately after contact with the prooxidant, and progressively increasing. Importantly, ROS production in mitochondria detected with MitoSox Red far preceded the appearance of the generalized oxidative stress in the yeast cell detected with H_2_DCF-DA (Figure 3 and Figure 4), which also increased exponentially, but only after a prolonged lag period. These findings seem to be not new at first, but they are new for fungal cells, and even more generally, for non-mammalian cells, thus reinforcing the notion that yeasts share well-conserved molecular and cellular mechanisms of eukaryotic cell biology. The results obtained were confirmed by additional experiments (Appendix A, Figure 5 and Figure 6). A significant increase in SODs activity coincided with the propagation of oxidative stress (Appendix A), while an increment in mitochondrial lipid peroxidation occurred predominantly before the development of the generalized oxidative stress (Appendix A).

Preincubation of yeast cells with the mitochondria-targeted antioxidant SkQ1 substantially diminished production of mROS (Figure 5), while only slightly alleviating the generalized oxidative stress (Figure 6).

This conclusion carries elements of novelty, as such a result could be expected, but had not yet been demonstrated. SkQ1 is a mitochondria-targeted antioxidant belonging to the SkQ family, where Sk means “penetrating cation” and “Skulachev ion”—a term put forward by D. Green [100], and Q refers to plastoquinone. It is one of the most intensively studied antioxidants, and it is effective in treating a large number of age-related pathologies [29,101,102,103]. It is pertinent to note that mitochondria-targeted lipophilic antioxidants offer advantages over conventional water-soluble antioxidants as they are transported and accumulated within cells and mitochondria in conformity with the membrane potential generated on the cytoplasmic or mitochondrial membrane, respectively. As a result, their concentrations in mitochondria would increase several orders of magnitude compared to the initial low nontoxic concentrations. Moreover, lipophilic antioxidants have a very high distribution coefficient between a membrane and a water phase, which contributes to further increase of their concentrations within mitochondria [29]. Even more astonishingly, the majority of mitochondria-targeted antioxidants of the SkQ family can be effectively recharged by the mitochondrial respiratory chain, thus ensuring their repetitive function. Finally, the accumulation of cationic mitochondria-addressed antioxidants within mitochondria at the expense of the membrane potential depolarizes the membrane (causing mild uncoupling), which in turn would prevent further cation entrance to the mitochondria, rescuing mitochondria from their damage.

The advantage of *D*. *magnusii* as a model organism allowed us to trace the relationship between the development of ROS production (both in mitochondria and in the whole cell) and mitochondrial structure. Mitochondria in most eukaryotes are highly dynamic organelles that continuously fuse and divide to adjust their shape according to different cell needs. These processes are orchestrated by the cellular machinery, which comprises dynamin-related proteins that are controlled by a variety of protein-protein interactions and posttranslational modifications [104]. In mammalian cells, Opa-1 (optic atrophy), Mitofusin-1, and Mitofusin-2 are the main profusion proteins, while a large GTPase Drp1 (dynamin-related protein) and its different receptors, including Mff (mitochondrial fission factor), Fis1(fission), and mitochondrial dynamic proteins MiD49 and MiD51, regulate mitochondrial fission [105]. Drp1 lacks a mitochondrial destination sequence, and multiple receptors are capable of recruiting Drp1 to mitochondria to induce fission [63]. In yeast (for example, *Saccharomyces cerevisiae*), Fzo1(fuzzyonions), a large GTPase belonging to the mitofusin family of proteins, joined with Mgm1 (mitochondrial genome maintenance), an ortholog of OPA1, are the main profusion proteins, while Dnm1 (dynamin-related), also a large GTPase, is recruited in the fission complexes with Fis1, Mdv1, and Caf4 at the mitochondrial outer membrane [106]. It has been argued that this kind of organellar dynamics is required to efficiently distribute mitochondria throughout the cell and generate highly interconnected networks that can facilitate efficient diffusive search processes [107] and play a key role in maintaining the integrity of the mitochondria. Importantly, excessive mitochondrial fission (fragmentation) is a prominent early event, contributing to cell death via mitochondrial dysfunction in the progression of various diseases [33,108,109,110].

Recently, we have shown that oxidative stress mediated by the prooxidant *t*-BHP triggered mitochondrial fragmentation in yeasts, and that mitochondria-targeted antioxidants of the SkQs family alleviated or even reversed this [77], making it possible that excessive mitochondrial fission could originate from mROS. Here, we confirmed these observations, showing that incubation with the prooxidant was accompanied not only by a gradual increase in fluorescence of MitoSox Red, which is indicative of mROS production, but also by a gradual fragmentation of mitochondria (Figure 4, Appendix A).

To establish closer correlations between mitochondrial fragmentation and the development of mitochondrial or generalized oxidative stresses (i.e., ROS detected by MitoSox Red or DCF, respectively) we used fluorescence microscopy. The increase in DCF fluorescence in all cells was found to occur strictly after the mitochondrial fragmentation peak had passed (Figure 7), suggesting that mitochondrial fragmentation was largely induced by mROS, and was an event preceding the development of generalized oxidative stress.

However, SkQ1 did not reduce fragmentation. It is pertinent to note that SkQ1 prevented or even reversed mitochondrial fragmentation in yeast cells [77], provided that the concentration of *t*-butyl hydroperoxide was 250 µM. In this study, we used 750 µM *t*-butyl hydroperoxide, as all experiments with MitoSox Red were restricted to the 80-min time period to avoid relocalization of MitoSox Red to the nucleus and the appearance of its fluorescence intensity not related to mitochondrial oxidative stress. We believe that SkQ1 could reduce fragmentation of mitochondria, presumably due to the irreversibility of the fragmentation process under these specified oxidative conditions. Further experiments are needed to better understand the regulation of processes involved in mitochondrial fragmentation, especially under different redox conditions.

In sum, we offered a new, highly promising model for studying the spatiotemporal propagation of oxidative stress in the cell, as well as a cross-talk between this stress and mitochondrial dynamics. For yeast cells, this is the first demonstration of this interplay. Our conclusion from the analysis of more than 100 individual cells was that mitochondrial fragmentation is largely induced by mROS, and precedes the development of generalized oxidative stress, which could be of general value.

## Figures and Tables

**Figure 1 antioxidants-10-00120-f001:**
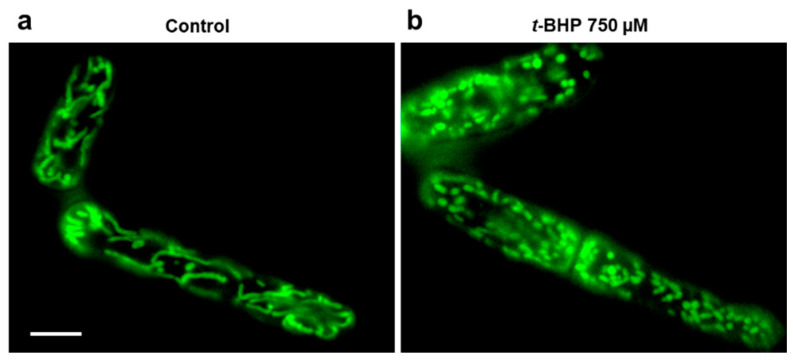
Visualization of mitochondria in exponentially grown *D. magnusii* cells. *t*-BHP-mediated fragmentation of mitochondria. *D. magnusii* cells washed with a fresh portion of growth medium were loaded with 200 nM Mitotracker Green for 30 min (**a**), then washed from the fluorescent probe and incubated at 250 µM *t*-BHP for 2 h (**b**). Stained cells were analyzed under a Zeiss Axioskop 40 fluorescence microscope. The scale bar is 10 µm.

**Figure 2 antioxidants-10-00120-f002:**
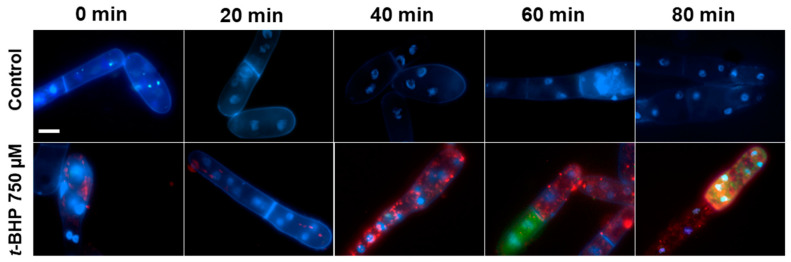
Localization of MitoSox Red, DCF, and Hoechst-33342 in the control *D. magnusii* cells (upper panel) and in cells under oxidative stress (bottom panel). Cells were harvested at the exponential growth phase, rinsed with 50 mM PBS, pH 5.5, and loaded with 15 µM H_2_DCF-DA (green fluorescence), 1 µM MitoSox Red (red fluorescence), and 5 µg/mL Hoechst-33342 (blue fluorescence) for 30 min; then washed in a fresh portion of incubation medium and exposed to 750 µM *t*-BHP. Stained cells were microscopically examined every 20 min. The scale bar is 10 µm.

**Figure 3 antioxidants-10-00120-f003:**
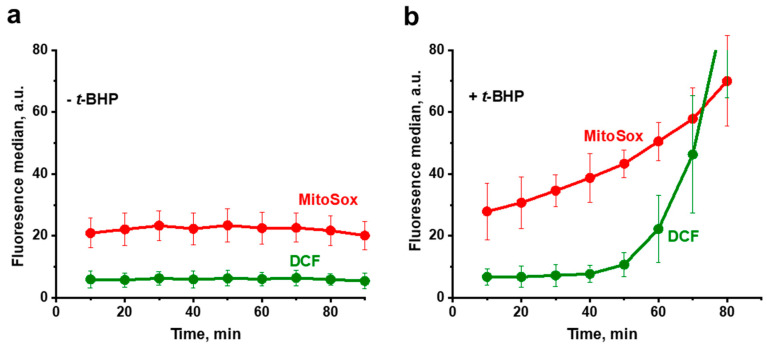
Propagation of *t*-BHP-induced ROS production in *D. magnusii* cells measured by flow cytometry. Cells were harvested at the exponential growth phase and rinsed with 50 mM PBS, pH 5.5; loaded with 15 µM H_2_DCF-DA (green fluorescence) and 1 µM MitoSox Red (red fluorescence) for 30 min; then washed in a fresh portion of incubation medium and exposed to 750 µM *t*-BHP. Cells were examined every 10 min. (**a**) Control cells not exposed to *t*-BHP. (**b**) Cells exposed to *t*-BHP. Data are presented as mean ± S.E. from three replicates.

**Figure 4 antioxidants-10-00120-f004:**
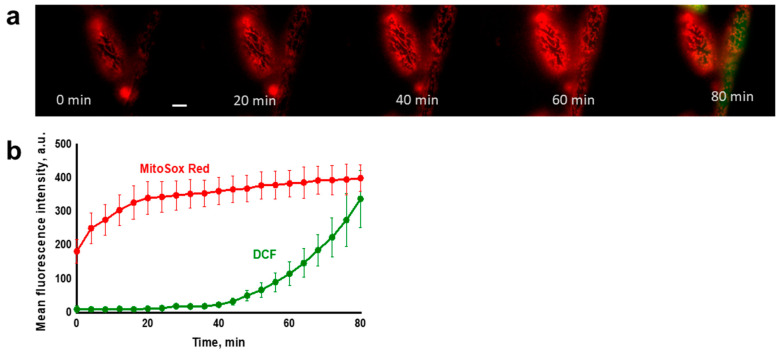
Propagation of *t*-BHP-induced ROS production in *D. magnusii* cells measured by time-lapse fluorescence imaging. Cells were treated as in Figure 3. Images were obtained every 3 min. (**a**) Cell images were selected after 20-, 40-, 60- and 80-min time intervals. The scale bar is 10 µm. (**b**) Dynamics of fluorescence intensity of MitoSox Red and DCF. Data are presented as mean ± S.E. from three replicates.

**Figure 5 antioxidants-10-00120-f005:**
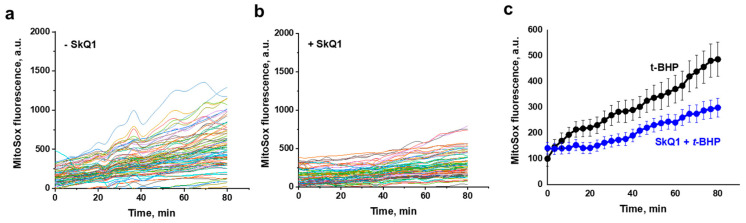
Effect of SkQ1 on *t*-BHP-induced mitochondrial ROS production (visualized by MitoSox Red) in *D. magnusii* cells measured using time-lapse fluorescence imaging. Cells harvested at the exponential growth phase were preincubated with 800 nM SkQ1 for 1 h, rinsed with 50 mM PBS, pH 5.5, loaded with 1 µM MitoSox Red and 15 µM H_2_DCF-DA for 30 min, then washed in a fresh portion of incubation medium and exposed to 750 µM *t*-BHP. (**a**)— Cells not treated with SkQ1; (**b**)—cells preincubated with SkQ1. (**a**,**b**)—MitoSox Red fluorescence intensity in individual cells presented; (**c**)—mean values of MitoSox Red fluorescence intensity; data are presented as mean and error bars from 101 replicates with 95% confidence interval.

**Figure 6 antioxidants-10-00120-f006:**
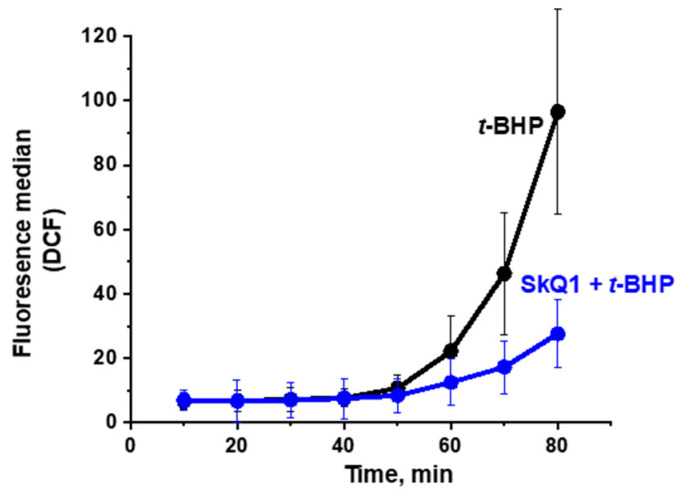
Effect of SkQ1 on *t*-BHP-induced ROS production (visualized by H_2_DCF-DA) in *D. magnusii* cells as measured by flow cytometry. Cells were treated as in Figure 3 and examined every 10 min. Data are presented as mean ± S.E. from three replicates.

**Figure 7 antioxidants-10-00120-f007:**
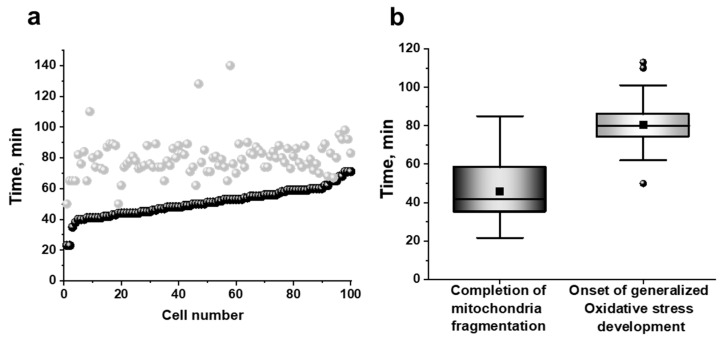
(**a**) The onset of the *t*-BHP-induced generalized oxidative stress (visualized by H_2_DCF-DA) (grey balls) and the completion of mitochondrial fragmentation (black balls) in 101 individual *D. magnusii* cells as measured using time-lapse fluorescence imaging. Cells were treated as in Figure 5 and examined every 3 min. (**b**) Summarizing presentation of the same results—the mean time periods required for initiating generalized oxidative stress and completing mitochondrial fragmentation in the same individual yeast cells.

## Data Availability

Data are contained within the article and Appendix A.

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
