# Peer review of "Propagation of Mitochondria-Derived Reactive Oxygen Species within the Dipodascus magnusii Cells"

_antioxidants, 2021, doi:10.3390/antiox10010120_

Round 1
Reviewer 1 Report
The manuscript by Rogov et al. entitled “Propagation of mitochondria-derived reactive oxygen species within the Dipodascus magnusii cells” examines the spatiotemporal development of prooxidant-induced oxidative stress in giant yeast cells. Overall, the manuscript is well-written with an informative introduction. As the authors use of D. magnusii as a model system permits relationship between the development of ROS production and structure to be examined, it is unclear what are the novel findings of this study. The authors discuss ROS-induced ROS release, mitochondrial fragmentation, and the scavenging of mROS, however a well-defined purpose/conclusion is not evident. A few specific comments are described below.
- The authors need to consider and address the question of “How does this study advances our current understanding of mitochondria ROS and oxidative stress? To strengthen the manuscript, the purpose/conclusions are not clear.
- In figure 1, the presence of the mitochondria is not evident. The authors should consider added a marker for the Mitochondria such as “MitoTracker” or a brightfield/phase contrast image as some fluorescent markers for the mitochondria can respond to changes in the its membrane potential.
- Legend for figures 2, 3b, 4, 5, and 6 should state the number of experiments and give some sense of statistical significance.
- Line 84, delete second “atrophy”
- Line 85, delete second “that”
- Lines 94/95 – repeated sentence
- Line 97, replace “from” with “of” in regard to the lattice-like, cell-wide network
- Line 313, the phase “to a laser extend on axons” appears out of place
Author Response
Response is enclosed

Reviewer 2 Report
This submission to investigate the ROS-induced ROS release (RIRR) in neighboring mitochondria using gigantic Dipodascus magnusii yeast cells. I like to give the following comments.
- In the abstract, conclusion seems unclear. Please improve it.
- Limitations regarding the research of RIRR carried out on cardiac myocytes need to introduce in detail. Then, the merits of yeast cell used in current study may indicate in clear.
- Source of yeast cells must show in clear. Amounts of cells used in each assay remained unknown. Indication of the suppliers for each instrument must follow the requirements of this journal.
- Concentration of the prooxidant t-BHP needs the reference(s) to support. The used antioxidant SkQ1 is fully unclear, including the source and reference(s).
- Figure 6 seems most important. However, it is hard to follow. Please revise it to be clear.
- The mitochondrial fragmentation was modified by SkQ1 or not? Please indicate it in clear.
- In conclusion, scavenging of mROS may inhibit excessive Drp1p-mediated mitochondrial fission. However, no data shown the changes in Drp1p-mediated mitochondrial fission.
- It seems better to show that similar changes were observed in cardiac myocytes or not.
Author Response
Response is enclosed

Reviewer 3 Report
This study investigated the spatiotemporal development of prooxidant-induced oxidative stress in a model of gigantic Dipodascus magnusii yeast cells. These cells contain a highly structured mitochondrial reticulum, and because of their size, they allow for the simultaneous assessment of ROS production and diffusion in mitochondria and whole cell through the use of fluorescence indicators specific for the two compartments. The data presented in the study indicate that the prooxidant-induced oxidative stress initiated in mitochondria preceded the appearance of generalized oxidative stress in the whole cell. Mitochondrial ROS were almost totally prevented by the use of the mitochondria-specific antioxidant SkQ1. Mitochondrial ROS production was accompanied by mitochondrial fragmentation, an event that occurred prior to the generalized oxidative stress development.
Comments:
The study addresses an important and still debated point in the field of ROS formation and propragation within the cells.
The methods used for the study are appropriate and the conclusion of the authors is supported by the data reported here.
A general question that should be commented in the discussion is whether the pattern observed in the yeast cells used here occurs in a similar manner in mammalian cells such as hepatocytes that can generate ROS in the endoplasmic reticulum through actuvation of specific cytochromes, in addition to generating ROS in the mitochondria. THis would be particualrly important in cells that present reduced mitochondria network, counterbalanced by mitochondria/endoplasmic reticulum contact points or interaction.
English style should be revised. A few sentences throughout the manuscript are ackwards at time to follow.
Author Response
Response is enclosed

Round 2
Reviewer 2 Report
It has been revised mostly according to comments. However, the mitochondrial fragmentation was modified by SkQ1 or not? It is an important background in the application of SkQ1. Additionally, indication of the suppliers for each instrument must show in clear. Both are required to add in 2nd round.
Author Response
Response is enclosed
